# Syringin: A Phenylpropanoid Glycoside Compound in *Cirsium brevicaule* A. GRAY Root Modulates Adipogenesis

**DOI:** 10.3390/molecules26061531

**Published:** 2021-03-11

**Authors:** Abu Yousuf Hossin, Masashi Inafuku, Kensaku Takara, Ruwani N. Nugara, Hirosuke Oku

**Affiliations:** 1The United Graduate School of Agricultural Sciences, Kagoshima University, Kagoshima 890-0065, Japan; yousuf.uoda@gmail.com (A.Y.H.); k-takara@agr.u-ryukyu.ac.jp (K.T.); okuhiros@comb.u-ryukyu.ac.jp (H.O.); 2Tropical Biosphere Research Center, University of the Ryukyus, Senbaru 1, Nishihara, Okinawa 903-0213, Japan; nilushinug@sjp.ac.lk; 3Faculty of Agriculture, University of the Ryukyus, Senbaru 1, Nishihara, Okinawa 903-0213, Japan; 4Faculty of Technology, University of Sri Jayewardenepura, Gangodawila, Nugegoda 10250, Sri Lanka

**Keywords:** *Cirsium brevicaule* A. GRAY, 3T3-L1 cells, anti-adipogenesis, syringin

## Abstract

*Cirsium brevicaule* A. GRAY is a wild perennial herb, and its roots (CbR) have traditionally been used as both food and medicine on the Japanese islands of Okinawa and Amami. The present study evaluated the antiadipogenic effect of CbR using mouse embryonic fibroblast cell line 3T3-L1 from JCRB cell bank. Dried CbR powder was serially extracted with solvents of various polarities, and these crude extracts were tested for antiadipogenic activity. Treatment with the methanol extract of CbR showed a significant suppression of lipid accumulation in 3T3-L1 cells. Methanol extract of CbR was then fractionated and subjected to further activity analyses. The phenylpropanoid glycosidic molecule syringin was identified as an active compound. Syringin dose dependently suppressed lipid accumulation of 3T3-L1 cells without cytotoxicity, and significantly reduced the expressions of peroxisome proliferator-activated receptor gamma, the master regulator of adipogenesis, and other differentiation markers. It was demonstrated that syringin effectively enhanced the phosphorylation of the AMP-activated protein kinase and acetyl-CoA carboxylase. These results indicate that syringin attenuates adipocyte differentiation, adipogenesis, and promotes lipid metabolism; thus, syringin may potentially serve as a therapeutic candidate for treatment of obesity.

## 1. Introduction

Obesity is a complex, chronic disease, with its adverse consequences reaching pandemic levels in recent times. Currently, more than 1.9 billion adults worldwide are overweight with over 650 million of them being obese. Notably, a significant increase in obese children under the age of five years reached 38 million in 2019 [1]. Obesity represents a major health challenge because it substantially increases risk of noncommunicable diseases, including cardiovascular diseases, cancer, and diabetes mellitus, which account for > 70% of early deaths worldwide, thus representing the leading cause of mortality and premature disability [2,3]. Obesity can be developed as a result of an increase in the number of adipocytes due to division and differentiation of preadipocytes, or storage of excess energy in the form of glycerol and fatty acids [4]. Adipocytes have a critical role in the regulation of adipose tissue and lipid homeostasis. Differentiation of preadipocytes involves the expression of genes and transcription factors, including CCAAT-enhancer-binding protein (C/EBP) α and peroxisome proliferator-activated receptor (PPAR) γ, which are responsible for the expression of adipogenesis-related genes [5]; during the late stages of differentiation, a momentous increase in lipogenic and lipogenesis-related genes is observed [6]. Inhibition of adipocyte differentiation from preadipocytes and the release of glycerol following the breakdown of triglycerides present in lipids are important for the prevention and management of obesity.

The genus *Cirsium* (thistle) belongs to the Asteraceae family; more than 200 species of thistle are distributed around the world. Some *Cirsium* species are used as edible and medicinal plants in traditional medicine, although other species are often considered as invasive weeds against which massive means of herbicidal control are deployed [7,8]. It is well known that *Cirsium* species contain a variety of natural products, and their main secondary metabolites include flavonoids and glycosides [9,10]. Many studies have reported that compounds from *Cirsium* species and their extracts show different biological activities, such as antioxidant, antidiabetic, anti-inflammatory, hepatoprotective, and anticancer activities [7].

It has been reported that flavones isolated from *C. japonicum* enhance adipocyte differentiation by inducing PPARγ activation in 3T3-L1 cells [11,12]. An antidiabetic effect of *C. japonicum* was also revealed in diabetic rats, suggesting their potential benefit as an alternative in treating diabetes mellitus [12]. Mori et al. demonstrated that the crude extracts of C. *oligophyllum* inhibited lipid accumulation in white adipose tissue in rats [13]. Nonpolar crude extracts of *C. pascuarense*, and *C. vulgare* and *C. ehrenbergii* have been shown to confer antidiabetic and hepatoprotective activities, respectively [14,15]. In China, the roots or entire plants of more than ten *Cirsium* species (*C. japonicum*, *C. eriophoroideum*, *C. escuientum*, *C. griseum*, *C. lineare*, *C. maackii*, *C. pendulum*, *C. setosum*, *C. souliei*, and *C. valassovianum*) have been used as a folk medicine for various diseases [10]. Previous studies have investigated the antioxidant activities of the methanol and water extracts of the roots of *C. japonicum*, and also confirmed that both extracts have high phenolic and flavonoid contents and beneficial activities against diabetes [16].

A wild perennial herb named *C. brevicaule* A. GRAY (CBAG) grows in rocky gravel or forest margins along maritime coastlines [17] and is native to Southern Japan and China. Stems, leaves, and roots of CBAG are traditionally used as both food and herbal medicine on the Japanese islands of Okinawa and Amami. Dietary intake of CBAG leaves (CbL) showed a significant decrease in hepatic lipid accumulation in mice fed a high-fat diet [18], and treatment with nonpolar crude extract of CbL significantly reduced cellular lipid accumulation in 3T3-L1 cells. By contrast, there are few scientific data on the biological activities of CBAG root (CbR). This study was aimed to characterize the anti-obesity properties of CbR in 3T3-L1 murine cell model. To the best of our knowledge, we are the first group to report on the anti-obesity properties of CbR.

## 2. Results

### 2.1. Effect of CbR Extracts Fractions on 3T3-L1 Cells

The methanol extract resulted in the highest extraction yield (5.5%, *w*/*w*), followed by the hexane extract (3.5%, *w*/*w*), water extract (1.5%, *w*/*w*), and chloroform extract (0.5%, *w*/*w*) of dried CbR powder (Figure 1A). Lipid accumulation in 3T3-L1 adipocytes was significantly decreased when treated with the chloroform extract (*p* ≤ 0.001) or methanol extract (*p =* 0.002) (Figure 1B). Further fractionation was carried out focusing on the methanol extract which had the highest yield among the active extracts. As shown in Figure 1A, the methanol extract was further fractionated into Fr-1 through Fr-5 via a silica gel open column. Among these five divided fractions, treatments with lower-polarity fractions Fr-1 (*p =* 0.008) and Fr-2 (*p =* 0.001) significantly inhibited lipid accumulation in 3T3-L1 cells (Figure 1C). Thus, Fr-1 and Fr-2 were profiled by HPLC (Appendix A) and Fr-1 was used for the further fractionation on the basis of low polarity.

### 2.2. Chemical Structure Identification of Active Compound from CbR

The active fraction Fr-1 was further fractioned through normal and reverse-phase high-performance liquid chromatography (HPLC) columns (Figure 2A). The purified active fraction was subjected to NMR analysis (Figure 2B). ^1^H-NMR (CDCl_3_, 400 MHz): δ 6.75 (s, H-2, H-4, 2H), 6.55 (d, *J* = 15.9 Hz, H-7), 6.33 (dt, *J* = 15.9 Hz, 5.6 Hz, H-8), 4.87 (overlapped with solvent signal, H-1′), 4.22 (dd, *J* = 5.5 Hz, 1.2 Hz, H-9, 2H), 3.86 (s, 3-OCH_3_ 5-OCH3, 6H), 3.81 (m, H-6′a), 3.70 (m, H-6′b), 3.50 (m, H-2′), 3.44 (m, H-4′), 3.43 (m, H-3′), 3.23 (m, H-5′). ^13^C-NMR (CDCl_3_, 100 MHz): δ 154.41 (C-3, -5), 135.92 (C-4), 135.32 (C-1), 131.32 (C-7), 130.07 (C-8), 105.48 (C-2, -5), 105.37 (C-1′), 77.88 (C-5′), 75.77 (C-2′), 71.38 (C-4′), 63.62 (C-9), 62.62 (C-6′), 57.05 (3-OCH_3_ 5-OCH_3_). ESI-MS analysis of the compound found the [M + Na]^+^ ion at *m*/*z* 395.2. With the interpretation of the spectral data, the chemical structure of the active component was identified as syringin (PubChem ID: 5316860, Figure 2C).

### 2.3. Effects of Syringin during Adipogenesis

To gain insight on the mechanism of the suppressive effect of syringin during the adipogenesis, the time course of the adipocyte differentiation in the presence of syringin was evaluated (Figure 3A). Cellular lipid accumulation was significantly inhibited (*p* ≤ 0.001 *and p =* 0.001) when syringin 20 µM was administered from day 0 to days 2 or 6 (Figure 3B). On the other hand, the inhibitory effects were not observed when 3T3-L1 cells were treated with syringin only for the latter period, between days 2 and 6 (*p =* 0.17) (Figure 3B).

### 2.4. Effects of Syringin on the Adipocyte Differentiation of 3T3-L1 Cells

The first 48 h of differentiation of 3T3-L1 cells has been generally considered as a critical window for the assessment of antiadipogenic effects [19]. To investigate the antiadipogenic effects of syringin, 3T3-L1 cells were treated with syringin from day 0 to day 2. Double-strand DNA (dsDNA) content in the cellular lysate from 3T3-L1 cells cultured with syringin was largely similar to that of the control cells (Figure 4A). Further, the cells were treated with different doses of syringin to investigate the dose dependency effect on 3T3-L1 cells and the intracellular lipid accumulation in mature adipocytes was visualized by Oil Red O staining as shown in (Figure 4C). Syringin treated groups decreased the amount of lipid in 3T3-L1 adipocytes. This observed reduction in lipid accumulation was confirmed by relative lipid quantification assay. As shown in (Figure 3B), 5, 10, and 20 μM syringin significantly decreased the lipid accumulation by 13%, 18%, and 29%, respectively. However, syringin at 2.5 μM showed no significant effect on the lipid accumulation.

To understand the molecular mechanisms related to syringin, the lipid metabolism-related genes in 3T3-L1 cells were evaluated. Treatment with syringin significantly decreased the mRNA levels of the lipogenic-related genes SCD1 (*p =* 0.004) and GLUT4 (*p =* 0.04), whereas it triggered the mRNA levels of FASN (*p =* 0.003) (Figure 4D). However, the protein levels of SCD1 were suppressed with no significance, while GLUT4 protein levels were significantly retarded as observed in its mRNA expression level. On the other hand, syringin significantly increased the expression levels of lipolysis-related genes LIPE (*p* ≤ 0.001) and CPT1a (*p* ≤ 0.001), and thermogenenic-related gene UCP2 (*p =* 0.04) (Figure 4E,F). The protein expression level of CPT1a and LIPE showed an increasing trend, however with no significance (Appendix A). The mRNA levels of lipolysis-related gene ACOX (*p* ≤ 0.001) were significantly decreased by syringin treatment (Figure 4E). Further, syringin significantly decreased the key adipogenesis-specific gene PPARγ (*p =* 0.002) and adipocyte marker genes FABP4 (*p =* 0.001) and ADIPOQ (*p* ≤ 0.001), compared with untreated 3T3-L1 cells (Figure 5A). The protein expression levels of adipogenesis-related proteins PPARγ (*p =* 0.04), FABP4 (*p =* 0.01), and C/EBPα (*p =* 0.04) were also significantly decreased in the presence of syringin (Figure 5B).

To elucidate the possible modulations in the signaling pathways related to adipogenesis, syringin was exposed to the cells for 48 h during the differentiation activation. Syringin upregulated (*p =* 0.005) the levels of phosphorylated AMPK and ACC (*p =* 0.005), while maintaining the total AMPK and ACC unchanged (Figure 6A,B). In addition, the ratio of phosphorylated Akt was significantly inhibited (*p =* 0.003) in 3T3-L1 cells treated with syringin (Figure 6C).

## 3. Discussion

This study demonstrated that methanol extract of CbR and its minimally polar contents significantly inhibited cellular lipid accumulation in 3T3-L1 adipocytes, as a precursor to assess the effects of CbR on the development of obesity (Figure 1). Further fractionation, identification of syringin as an active compound in CbR (Figure 2), and elucidation of the ability of syringin to affect the lipid metabolism and adipogenesis-related genes at the transcriptional and posttranscriptional levels (Figure 4, Figure 5 and Figure 6) suggested its potential antiadipogenic activities.

Syringin is a phenylpropanoid glycoside, also called eleutheroside B. This compound was first isolated from the lilac *Syringa vulgaris*, and is distributed widely throughout several continents [20]. Many studies have reported beneficial activities of syringin, such as hypoglycemic, anti-inflammatory, antioxidative, and immunomodulating effects [21,22,23]. The potential hypoglycemic and antidiabetic activities of syringin are especially promising [24,25]. Syringin was also isolated from the roots of a different plant, *C. japonicum* [26]. Few studies, however, have investigated the effects of syringin on 3T3-L1 adipocytes and adipose tissues [27]. Syringin dose-dependently attenuated adiposity in 3T3-L1 cells at 5 μM, however, with least effect observed during the late stage of adipogenesis (days 2–6). This is the first report to show the antiadipogenic effect of syringin in the early stage of differentiation of 3T3-L1 cells (Figure 3B and Figure 4B,C) without cytotoxicity (Figure 4A).

There are several strategies for reducing obesity, such as lipogenic reduction and lipolytic enhancement [28]. Adipocytes play a major role in glucose and lipid metabolism, and respond to physiological signals or metabolic stresses by releasing endocrine factors that regulate diverse processes, such as energy expenditure, appetite control, glucose homeostasis, insulin sensitivity, inflammation, and tissue repair [29]. Adipocyte differentiation and fat accumulation are positively correlated with the number and size of adipocytes, which is associated with the development of obesity [30,31]. Treatment of 3T3-L1 cells with syringin significantly decreased the mRNA levels of major lipogenesis-related genes and enhanced mRNA levels of major lipolysis and thermogenesis-related genes (Figure 4D–F). Adipocyte differentiation is associated with a broad transcriptional network mediated by several transcription factors, which are responsible for expression of adipogenesis-related proteins. Among these transcription factors, PPARγ and C/EBP are major regulators of adipogenesis [32,33]. As shown in Figure 5, syringin attenuated the mRNA levels of adipocyte-specific genes such as FABP4 and ADIPOQ, which are regulated by C/EBPα or PPARγ during adipocyte differentiation [34]. Indeed, syringin suppressed the protein expressions of PPARγ and C/EBPα (Figure 5) and reduced cellular lipid accumulation during early adipocyte differentiation (Figure 3). These data suggest that syringin inhibited adipogenesis in 3T3-L1 preadipocytes at an early stage by modulating the expression of adipocyte differentiation-related transcription factors.

Recent studies have demonstrated the protective effects of syringin against oxidative stress and inflammation in diabetic rats, and suggested that syringin may be a potential candidate for the treatment of gestational diabetes [35]. It was also shown that syringin improved insulin sensitivity by increasing AMP-activated protein kinase (AMPK) activity, decreasing expression of lipogenic genes in skeletal muscle cells, and suppressing the chronic inflammation and endoplasmic reticulum (ER) stress [36]. AMPK, a serine/threonine protein kinase, regulates fatty acid synthesis and degradation, acts as a sensor of cellular energy metabolism, and is critical in maintaining energy homeostasis in the whole body [37]. It is known that AMPK plays a crucial role in regulating adipogenesis. The AMPK activation in adipose tissue could prove beneficial in attenuating dysfunction in adipose tissue, as it has a vital role in the regulation of transcriptional factors related to adipogenesis and lipid synthesis [38,39,40]. Therefore, an inhibitor of AMPK signaling or its downstream substrate activation that prevents the differentiation of preadipocytes into adipocytes could be considered a potential treatment of obesity [41,42,43]. In an attempt to elucidate the molecular mechanisms underlying syringin-induced anti-adipogenesis of 3T3-L1 preadipocytes, the protein levels of phosphorylated AMPK and its substrate, ACC, were investigated. Treatment with syringin increased the levels of p-AMPK (Thr172), and this activation resulted in the phosphorylation of ACC, which in turn inhibits adipogenesis with the inhibition of ACC (Figure 6A,B). The present study also showed decreases in lipogenesis-related gene expression (Figure 4D), suggesting that syringin effectively attenuates adipogenesis via the regulation of AMPK activation in preadipocytes. A significant decrease in the p-Akt/Akt ratio was observed in syringin-treated cells compared to that of the control (Figure 6C); it has been reported that inhibition of Akt phosphorylation/activation blocks adipocyte differentiation of 3T3-L1 cells, and suggests syringin’s ability to inhibit the Akt activation [44,45]. On the other hand, syringin treatment did not affect the C/EBPα mRNA levels, although a significant decrease in their protein levels was observed (Figure 5). It has been known that C/EBPα translation is inhibited by endogenous molecules, which relate to ameliorating ER stress [46,47]. It is noteworthy that the expression levels of protein and mRNA showed a discrepancy in the presence of syringin, however, leading to a significant attenuation of adipogenesis. Further investigations on the molecular characterization of syringin will be beneficial to understanding the underlying mechanisms.

## 4. Materials and Methods

### 4.1. Isolation and Identification of Active Compounds from CbR

*Cirsium brevicaule* A. Gray was botanically classified by Asa Gray [48]; a voucher specimen of the plant was deposited in the Amami Museum (Kagoshima, Japan) by Hayao Ohno (voucher specimen number: 1582). The CbR used in this study was harvested on Tokunoshima Island in Kagoshima Prefecture, Japan. Freeze-dried CbR was generously provided by Healthy Island Co. (Kagoshima, Japan). Dried powder was serially extracted by incubation with a total of 9 volumes of hexane, chloroform, methanol, and water for 1 h at 37 °C. To identify the antiadipogenic compounds, the methanol extract was dried and re-dissolved in chloroform:methanol (9:1, *v*/*v*), and the soluble fraction was applied to a Hi-FLASH silica gel open-column (Yamazen Corp., Osaka, Japan) (Figure 1A). The starting eluent was chloroform:methanol (9:1, *v*/*v*) followed by a gradual shift in the mixing ration to 0:10 (*v*/*v*) to yield 5 fractions (Fr-1 to Fr-5). Fr-1 was further subjected to HPLC with (chloroform:methanol, 9:1, *v*/*v*) in a silica gel column (Cosmosil 5SL-II, Nacalai Tesque, Inc., Kyoto, Japan). The active fractions were evaporated to a complete dry, dissolved in methanol, and further purified through an HPLC reverse-phase C18 column (GL Sciences Inc., Tokyo, Japan) with methanol containing 0.1% formic acid. All extracts and isolated fractions were evaporated or freeze-dried in vacuo, and stored at −80 °C until further use. Powders were dissolved in dimethyl sulfoxide before use for the treatments. To identify the chemical structure of the purified active compound, NMR spectra were measured on a Bruker AVANCE 400 (Bruker Biospin, Rheinstetten, Germany). ^1^H-NMR, ^13^C-NMR, HSQC, and HMBC were measured using a 5-mm probe. The operating frequencies were 400.13 MHz for ^1^H-NMR and 100.62 MHz for ^13^C-NMR spectra. Samples were measured at 299 K in CDCl_3_ with TMS as standard. ESI-MS was also performed on a LC-MS (Xevo TQD, Waters Corp., Milford, MA, USA) in positive-ion mode.

### 4.2. Cell Culture and Treatments

Dulbecco’s modified Eagle’s medium (DMEM) and human insulin were purchased from FUJIFILM Wako Pure Chemical Corporation (Osaka, Japan). Fetal bovine serum (FBS) and newborn calf serum (NCS) were purchased from AusGeneX PTY Ltd. (Oxenford, Australia) and Global Life Science Technologies Japan Co. Ltd. (Tokyo, Japan), respectively, and the sera were inactivated at 56 °C for 30 min before use. Dexamethasone, 3-isobutyl-1-methylxanthine, and syringin were purchased from Sigma-Aldrich, Inc. (St. Louis, MO, USA). Cells were purchased from the JCRB Cell Bank (Tokyo, Japan) and cultured in a humidified atmosphere of 95% air and 5% CO_2_ at 37 °C.

3T3-L1 preadipocytes were maintained in DMEM with low glucose (1 g/L) containing 10% NCS, and complete confluence was avoided prior to initiating differentiation. Preadipocytes were cultured in 24-well plates at a density of 4 × 10^4^ cells per well. Confluent preadipocytes were maintained for 2 days prior to induction of differentiation by the following standard inducers: 0.5 mM 3-isobutyl-1-methylxanthine (IBMX), 0.25 µM dexamethasone (DEX), and 10 µg/mL insulin in DMEM with high glucose (4.5 g/L) containing 10% FBS for 48 h from days 0 to 2. The culture medium was then changed to DMEM with high glucose supplemented with 10% FBS and 10 µg/mL insulin from days 2 to 6. Cells were cultured in media supplemented with each extract or isolated fraction from days 0 to 6. To study at which stage the active compound functioned in adipocyte development, cell cultures were treated with syringin at different time intervals. On day 6, lipid droplets of 3T3-L1 cells were stained with Oil Red O using a standard protocol, and the stained dye was extracted for quantification by absorbance readout at 520 nm. To elucidate cytotoxic effects of syringin on 3T3-L1 cells, cells were treated with 20 μM syringin from day 0 to day 6. At the end of the treatment period, cells were washed twice with PBS and lysed in 1% Triton X-100. The double-strand DNA (dsDNA) content of cell lysates was determined using a Quanti-iT PicoGreen dsDNA Assay Kit (Thermo Fisher Scientific, Waltham, MA, USA).

### 4.3. Quantitative Real-Time Polymerase Chain Reaction

3T3-L1 cells were treated with 20 µM syringin or DMSO for 48 h (days 0 to 2), after which total RNA was extracted using the FastGene RNA Basic kit (NIPPON Genetics Co., Ltd., Tokyo, Japan). First-strand cDNA was synthesized using mRNA as a template. For quantitative real-time polymerase chain reaction (qRT-PCR), the primers and probe sets for acetyl-coenzymeA (CoA) carboxylase 1 (ACC1, Mm.PT.58.12492865), Adiponectin (ADIPOQ, Mm.PT.58.9719546), β-actin (ACTB, Mm.PT.58.33257376.gs), C/EBPα (Mm.PT.58.30061639.g), carnitine palmitoyltransferase 1A (CPT1a, Mm.PT.58.10147164), carnitine palmitoyltransferase 2 (CPT2, Mm.PT.58.13124655), fatty acid binding protein 4 (FABP4, Mm.PT.58.43866459), fatty acid synthase (FASN, Mm.PT.58.14276063), hormone-sensitive lipase (LIPE, m.PT.58.6342082), lipoprotein lipase (LPL, Mm.PT.58.46006099), peroxisomal acyl-coenzyme A oxidase 1 (ACOX, Mm.PT.58.50503784), PPARγ (Mm.PT.58.31161924), PPARγ coactivator 1α (PGC1a, Mm.PT.58.16192665), stearoyl-CoA desaturase-1 (SCD1, Mm.PT.58.8351960), glucose transporter type 4 (GLUT4, Mm.PT.58.9683859), UCP2 (Mm.PT.58.11226903), and UCP3 (Mm.PT.58.9090376) were purchased from Integrated DNA Technologies, Inc. (Coralville, IA, USA). To measure the relative abundance of target transcripts, amplifications were performed using PrimeTime Gene Expression Master Mix (Integrated DNA Technologies, Inc.) in the StepOne real-time PCR system (Thermo Fisher Scientific), and the amounts of the target transcripts were normalized to those of ACTB.

### 4.4. Western Blot Analyses

Cultured 3T3-L1 cells treated with 20 µM syringin from days 0 to 2 were washed with PBS, and then lysates were prepared with PRO-PREP™ protein extraction solution (iNtRON Biotechnology, Gyeonggi-do, Korea). Proteins were extracted and separated by SDS-PAGE and subsequently transferred onto polyvinylidene difluoride (PVDF) membranes (GE Healthcare Life Sciences, Chalfont, UK). Membranes were blocked with ATTO EzBlock Chemi (ATTO Corp., Tokyo, Japan) at room temperature for 1 h, and then reacted with the following monoclonal antibodies (mAb) or polyclonal antibodies (pAb): anti-GLUT4 mAb (1F8), anti-SCD1 pAb (M38), anti-LIPE pAb (D6W5S), anti-PPARγ mAb (81B8), anti-C/EBPα mAb (D56F10), anti-FABP4 mAb (D25B3), anti-ACC pAb, anti-p-ACC (Ser79) pAb, anti-Akt mAb (C67E7), anti-p-Akt (Ser473), mAb (D9E), and anti-β-actin mAb (E135) as primary antibodies, and anti-rabbit and anti-mouse IgG polyclonal secondary antibodies conjugated with horseradish peroxidase (HRP) purchased from Cell Signaling Technology, Inc. (Danvers, MA, USA), anti-AMPKα1/2 pAb and anti-p-AMPK1/2α (Thr172) pAb purchased from Santa Cruz Biotechnology, Inc. (Heidelberg, Germany), and anti-CPT1a mAb (8F6AE9) purchased from Abcam (Cambridge, UK). These antibodies were diluted with Bullet ImmunoReaction Buffer (Nacalai Tesque) and membranes were incubated for 1 h at room temperature. After each reaction, the PVDF membrane was washed three times with tris-buffered saline with tween 20 (TBST), and the protein bands were then detected using the ECL Advance Western Blotting Detection System (GE Healthcare). Images were captured and visualized using ImageQuant LAS 4000 (GE Healthcare), and the band intensities were quantified with ImageJ software. Band density was normalized to the level of β-actin.

### 4.5. Statistical Analyses

Data are expressed as mean ± standard error of the mean (SEM). Statistical significance of the difference between two experimental groups was determined using the Student’s t-test. To determine the significance of the differences among the means for more than three groups, the data were analyzed using one-way analysis of variance, and the differences among the mean values were subsequently evaluated using the Dunnett’s significant difference test. The level of significance was set to *p* < 0.05.

## 5. Conclusions

The present study demonstrated that CbR has potent anti-obesity activity partly via the attenuation of adipocyte differentiation. It was unveiled that syringin isolated from CbR attenuates lipogenesis by direct modulation of lipogenic gene expression and significantly inhibited adipogenesis of 3T3-L1 cells during the early stage of differentiation. Although additional studies are needed to elucidate the underlying molecular mechanisms and reveal other active components in CbR, our data provide evidence that CbR could be used in the treatment of obesity and diabetes in the future.

## Figures and Tables

**Figure 1 molecules-26-01531-f001:**
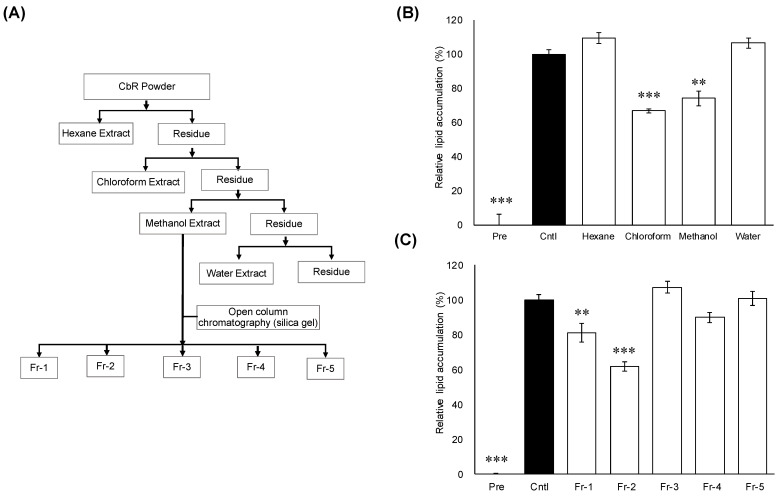
Effects of extracts and fractions from *Cirsium brevicaule* A GRAY root (CbR) on 3T3-L1 cells. (**A**) Flowchart for the isolation of active anti-obesity compounds from *Cirsium brevicaule* A GRAY root (CbR). (**B**) Inhibitory effects of the CbR extract on cellular lipid accumulation. (**C**) Inhibitory effects of the fractions derived from the methanol extract on cellular lipid accumulation. 3T3-L1 cells were treated with 500 μg/mL of crude extracts or 100 μg/mL of individual fractions of the crude extracts with 0.5% DMSO as the vehicle. The results are presented as means ± SEM of three independent experiments (*n =* 3). The asterisk (*) indicates a significant difference between control and treatment groups by Dunnett’s test. ** *p* < 0.01, and *** *p* < 0.001 vs. control (Cntl) and preadipocytes (pre).

**Figure 2 molecules-26-01531-f002:**
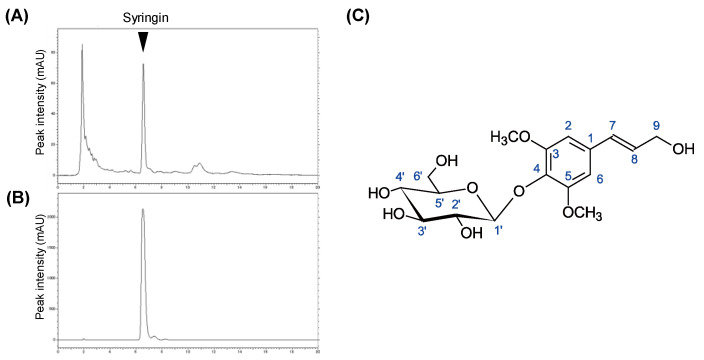
Isolation of active components from CbR and chemical structure identification. (**A**,**B**) Chromatograms of Fr-1 from crude methanol extract and purified syringin, respectively. (**C**) Chemical structure of syringin revealed by NMR analyses.

**Figure 3 molecules-26-01531-f003:**
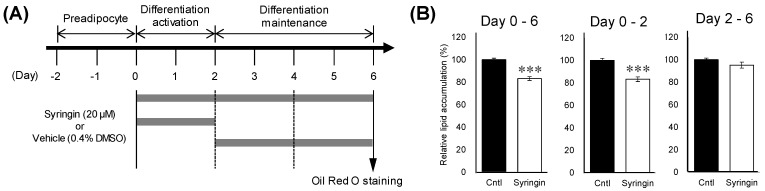
Effect of syringin on intercellular lipid accumulation in 3T3-L1 cells. (**A**) Time course of treatment with syringin during the cellular differentiation of 3T3-L1 adipocytes. (**B**) Inhibitory effects of syringin on lipid accumulation in 3T3-L1 adipocytes. The results are presented as means ± SEM of three independent experiments (*n* = 3). The asterisk (*) indicates a significant difference between control and treatment groups by the Student’s *t*-test. *** *p* < 0.001 vs. control (Cntl).

**Figure 4 molecules-26-01531-f004:**
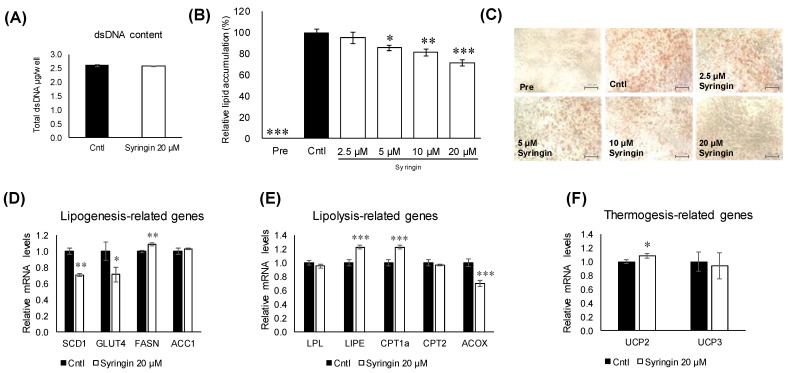
Effect of syringin on adipogenic differentiation through the regulation of adipogenic factors in 3T3-L1 cells. (**A**) Cytotoxic effect of syringin on 3T3-L1 cells. (**B**,**C**) Cellular lipid accumulation by Oil Red O staining (image 20×magnification and scale bar = 100 µm). (**D**–**F**) mRNA levels of (**D**) lipogenesis-, (**E**) lipolysis-, and (**F**) thermogenesis-related genes. Experiments were performed in triplicate. The results are presented as means ± SEM of three independent experiments (*n =* 3). The asterisk (*) indicates a significant difference between control and treatment groups by Dunnett’s test and Student’s t-test. * *p* < 0.05, ** *p* < 0.01, and *** *p* < 0.001 vs. control (Cntl) and preadipocytes (pre).

**Figure 5 molecules-26-01531-f005:**
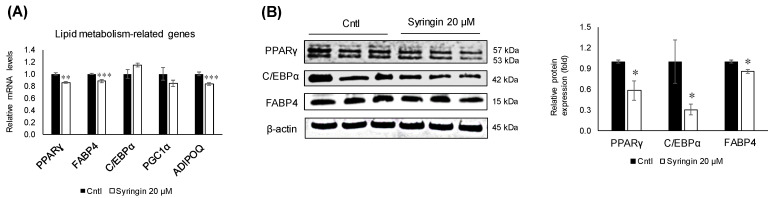
Effect of syringin on the expression of adipogenesis-related genes and proteins. (**A**) mRNA levels of adipogenesis-related genes. (**B**) Representative Western blot of adipogenic protein factors. The results are presented as means ± SEM of three independent experiments (*n =* 3). The asterisk (*) indicates a significant difference between control and treatment groups tested by Student’s t-test. * *p* < 0.05, ** *p* < 0.01, and *** *p* < 0.001 vs. control (Cntl).

**Figure 6 molecules-26-01531-f006:**
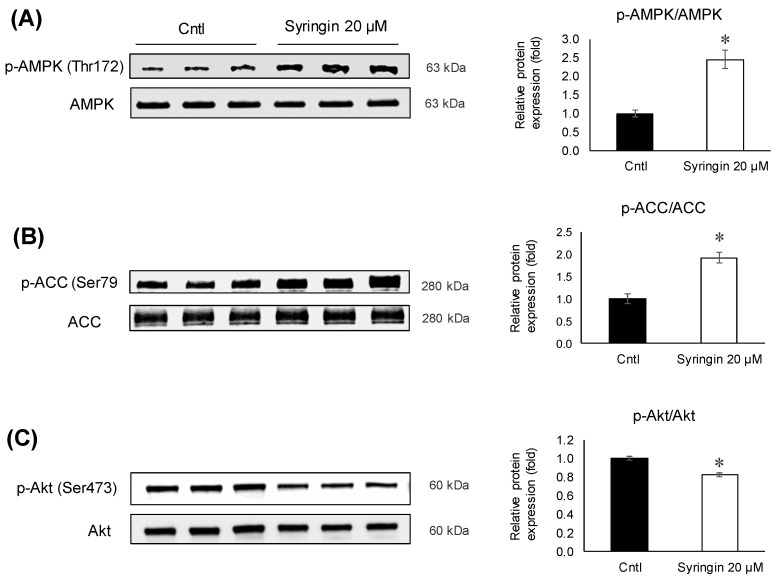
Effects of syringin on AMPK, ACC, and Akt phosphorylation during the differentiation of 3T3-L1 adipocytes. (**A**–**C**) Ratios of relative phosphorylation to total protein levels of (**A**) p-AMPK/AMPK, (**B**) p-ACC/ACC, and (**C**) p-Akt/Akt. The results are presented as means ± SEM of three independent experiments (*n =* 3). The asterisk (*) indicates a significant difference between control and treatment groups by Student’s t-test. * *p* < 0.05 vs. control (Cntl).

## Data Availability

The data presented in this study are available in this article.

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
