# Peer review of "Syringin: A Phenylpropanoid Glycoside Compound in Cirsium brevicaule A. GRAY Root Modulates Adipogenesis"

_molecules, 2021, doi:10.3390/molecules26061531_

Round 1

Reviewer 1 Report

The manuscript entitled "Syringin: A Phenylpropanoid Glycoside Compound in Cirsium brevicaule A. GRAY Root Modulates Adipogenesis" described the antiadipogenic effect of Cirsium brevicaule roots four solvents with  various polarities using 3T3-L1 cells. It is very interesting scientific work and very important due to the increase of obesity rates globally. The manuscript needs major corrections as:

  1. In the Abstract section: write kindly the full name and source of 3T3-L1 cells.
  2. Remove"(PubChem ID: 5316860)" from the abstract and write it in the Results section.
  3. Write shortly the meaning of AMP-activated protein kinase (AMPK) is a sensor of cellular energy status that regulates cellular and whole-body energy balance.
  4. Kindly write briefly the last statistics of obesity by WHO in the Introduction section.
  5. Line 70 start the sentence with a full name of the plant
  6. Avoid the use of (We or Our) throughout the whole manuscript.
  7. In section "4.1. Isolation and identification of active compounds from CbR" Kindly write the name of botanist who classified the plant, in which department the voucher specimen code was conducted and also provide the voucher specimen code for the screened plant.
  8. In the front of "Wako Pure Chemical Industries Ltd." write its location.
  9. Avoid starting any sentence with abbreviations.
  10. In the results section add the p values for all the obtained results.
  11. Rewrite the conclusion section to be clear for readers also remove the citations from it
  12. The manuscript needs grammars, typos and editing corrections before been submitted

Reviewer 2 Report

Dear Editor,

The manuscript submitted by Dr Hossin and collaborators aims at investigating the biological activities of herbal extracts (and its fractions) obtained from the roots of Cirsium brevicaule A. GRAY (CbR) is a wild perennial herb, in pre-adipocyte cell lines (3T3-L1 cells).

The studies reveals that the main active constituent is syringin, a phenylpropanoid glycosidic molecule.

In vitro work in 3T3-L1 cells showed that both CbR and syringin suppressed lipid accumulation, mainly by hindering the activity of PPAR gamma, a master regulator of adipogenesis and several adipocyte differentiation markers. The authors also find that syringin successfully enhanced lipid metabolism.

In general, this in vitro work is interesting and deserved merit, as it introduces a novel compound that may exert significant anti-obesogenic activities, despite further in vivo work may be warranted in the future.

Nonetheless, there are some issues that require the authors attention. These are listed below as major and minor comments:

Major comments:

  1. Unless there is preliminary evidence that the concentrations and exposure times are suitable to elicit a significant anti-adipogenic response in differentiated 3T3-L1 cells, dose-curve and time-course analyses of CbR, its fractions and of syringin should also be provided.
  2. It should be noticed that DMSO has proven to significantly reduce metabolic activities, trigger apoptosis and increase oxidative stress at concentrations as low as 0.1% in the oxidative stress-sensitive 3T3-L1 cell lines and to hinder lipid accumulation at concentrations above 10% (Dludla, P. V., Jack, B., Viraragavan, A., Pheiffer, C., Johnson, R., Louw, J., & Muller, C. (2018). A dose-dependent effect of dimethyl sulfoxide on lipid content, cell viability and oxidative stress in 3T3-L1 adipocytes. Toxicology reports, 5, 1014–1020). Given that DMSO affects cellular metabolism already at low concentrations, it is recommended that a non-DMSO control is added to ascertain that the vehicle itself is not partially masking any beneficial effects of the tested compound.
  3. It was surprising to see that most lipolysis- (Figure 4E) and lipid-metabolism related genes (Figure 5A) were only marginally affected (0.1-0.2 fold change) whereas the biological effects (Figure 4B and 4C) and changes in protein expression (Figure 5B and 5C) were quite robust. Given the different kinetics of mRNA Vs protein expression and subsequent biological effects are different, it is likely that gene expression data does not provide a realistic snapshot of how syringin affects gene expression. This reviewer recommends the author to add a paragraph in the discussion to highlight these observations.

 Minor comments:

  1. Figure 4 (Panel C) – Scale bar is missing in photomicrographs. Please also add in the related caption.
  2. Figure 5 (Panel B) and Figure 5 (Panels A-C) – Please add MW next to the Western blot bands shown.
  3. N-values to indicate how many biological samples where used should be indicated in the captions to figures.

Reviewer 3 Report

This article “Syringin: A Phenylpropanoid Glycoside Compound in Cirsium brevicaule A. GRAY Root Modulates Adipogenesis”. The comments for this manuscript are as follows:

  1. In Fig. 1 of Supplementary data, it is not indicated which is Fr-1 and Fr-2, please indicate clearly.
  2. The format of the references in this manuscript is very messy. The every first word in the titles of the cited references should not be capitalized. The format of the references should be written in accordance with the journal's regulations. At the same time, the format of the cited references should be unified. The author’s initials should also be capitalized. Examples are references: 4, 11, 12, 14, 15, 17, 18, 19, 20, 21, 22, 26, 28, 29, 30, 32, 36, 37, 39, 41, 42, 43 and 44 . There are errors in these references, please correct them.
  3. Line 113 is missing a period. In the description of line 96, Figure 1, “Cirsium brevicaule” should be italicized because it is a scientific name, and the word “J” from line 106 to 107 also needs to be italicized. In addition, the resolution of figure 1 is too poor, can it be placed on a clearer figure?
  4. The author shows the mRNA expression level of many proteins in Figure 4, why do not perform Western blot further to see if the protein expression level is consistent with the mRNA expression level. After all, the performance of protein is the data that can persuade readers. mRNA expression just represents this potential, but it does not necessarily show protein.
  5. The explanation of figure 4C should be a bit more so that readers can understand the significance of the experimental results. But the authors only mentioned it in line 205. If it is not important, you don't need to put it in. Since it is put in figure 4C, there should be a detailed explanation, otherwise the reader can't seem to see the difference.

My suggestion is major revision.

Round 2

Reviewer 1 Report

The authors established all the required corrections suggested from me

Author Response

We greatly and deeply appreciate your cooperation and consideration to develop the review process in our manuscript (molecules-1101855).

Reviewer 2 Report

Dear Editor,

The authors have provided reasonable explanations to address most of the issues raised by this reviewer, so there are no further comments or concerns over the quality/content of the work submitted to Molecules.

Author Response

(The authors gave the same response as above.)

Reviewer 3 Report

The manuscript is improved significantly. But I still feel dissatisfied with the response for the fourth question. The authors do not respond to the reviewer's question. The content of their reply is suspected of evading the problem. I think the authors should do more experiments to answer this question.
